# Neuroprotective Effects of a Hydrogen Sulfide Donor in Streptozotocin-Induced Diabetic Rats

**DOI:** 10.3390/ijms242316650

**Published:** 2023-11-23

**Authors:** Abdulaziz M. F. Shayea, Waleed M. Renno, Bedoor Qabazard, Willias Masocha

**Affiliations:** 1Department of Occupational Therapy, College of Allied Health Science, Kuwait University, P.O. Box 24923, Safat 13110, Kuwait; abdulazez.shayee@ku.edu.kw; 2Molecular Biology Program, College of Graduate Studies, Kuwait University, P.O. Box 24923, Safat 13110, Kuwait; 3Department of Anatomy, College of Medicine, Kuwait University, P.O. Box 24923, Safat 13110, Kuwait; waleed.renno@ku.edu.kw; 4Department of Pharmacology and Therapeutics, College of Pharmacy, Kuwait University, P.O. Box 24923, Safat 13110, Kuwait; bedoor.qabazard@ku.edu.kw

**Keywords:** hydrogen sulfide, diabetic neuropathy, apoptosis, neuroprotective

## Abstract

Diabetic neuropathy is an important long-term complication of diabetes. This study explored the hypothesis that hydrogen sulfide (H_2_S) ameliorates neuropathic pain by controlling antiapoptotic and pro-apoptotic processes. The effects of a slow-releasing H_2_S donor, GYY4137, on the expression of antiapoptotic and pro-apoptotic genes and proteins, such as B-cell lymphoma 2 (Bcl2) and Bcl-2-like protein 4 (Bax), as well as caspases, cyclooxygenase (COX)-1 and COX-2, monocytes/macrophages, and endothelial cells, in the spinal cord of male Sprague-Dawley rats with streptozotocin-induced peripheral diabetic neuropathy, were investigated using reverse transcription-PCR, western blot and immunohistochemistry. The antihypoalgesic activities of GYY4137 on diabetic rats were evaluated using the tail flick test. Treatment of diabetic rats with GYY4137 attenuated thermal hypoalgesia and prevented both the diabetes-induced increase in *Bax* mRNA expression (*p* = 0.0032) and the diabetes-induced decrease in *Bcl2* mRNA expression (*p* = 0.028). The GYY4137-treated diabetic group had increased COX-1 (*p* = 0.015), decreased COX-2 (*p* = 0.002), reduced caspase-7 and caspase-9 protein expression (*p* < 0.05), and lower numbers of endothelial and monocyte/macrophage cells (*p* < 0.05) compared to the non-treated diabetic group. In summary, the current study demonstrated the protective properties of H_2_S, which prevented the development of neuropathy related behavior, and suppressed apoptosis activation pathways and inflammation in the spinal cord. H_2_S-releasing drugs could be considered as possible treatment options of diabetic peripheral neuropathy.

## 1. Introduction

Diabetes mellitus (DM) is one of the several major causes of the global burden of disease. It has several complications, which result in an increased healthcare burden. Diabetes related costs are estimated to be more than $7000 per person annually [1,2]. The complications of diabetes are microvascular and macrovascular. Macrovascular complications include atherosclerosis, which is common in patients with DM. The microvascular complications, which include diabetic nephropathy, neuropathy, and retinopathy, originate from chronic hyperglycemia. Diabetes is a common cause for neuropathy in patients after all other causes of neuropathy are excluded [3]. Oxidative stress causes accumulation of free radicals and reduced activity of antioxidant enzymes, which are associated with development of diabetic neuropathy. In animal models of diabetes, antioxidant treatment ameliorated oxidative stress and alleviated symptoms of diabetic neuropathy [4]. Hyperglycemia leads to increased oxidative stress which plays a pivotal role in the development of diabetic neuropathy by damaging the cells including endothelial, retinal, mesangial, and neural cells. Impaired glucose metabolism in diabetic condition causes an accumulation of glucose and glycolytic intermediates, which, instead of utilizing the glycolysis pathway is shunted to other metabolic or non-metabolic pathways, resulting in activation of the polyol pathway, hexosamine pathway, and advanced glycation end-products (AGEs) and protein kinase (PKC) pathway, which are involved in the pathogenesis of diabetes complications including painful neuropathy [4,5].

Diabetic neuropathy is a sign and symptom of peripheral nerve dysfunction in patients with DM. About 50% of diabetic patients suffer from neuropathy [3]. In type one diabetes mellitus (T1DM), both islet dysfunction and peripheral insulin resistance are present, and both are necessary for the development of hyperglycemia, leading to neural damage [6]. This damage leads to neuropathic pain, which can manifest as allodynia, hyperalgesia, hypoalgesia, paroxysmal pain, numbness, and temporal summation [7]. Allodynia is defined as pain caused by a stimulus that normally does not provoke pain. [8]. In contrast, hyperalgesia is defined as increased sensitivity to pain, which is caused by a stimulus that normally provokes pain. In addition, one of the complications of T1DM is the dysfunction of endothelial cells, which can occur either in diabetic neuropathy or diabetic retinopathy [9,10].

Apoptosis, also known as programmed cell death, is a mechanism for maintaining cell populations in tissues and may occur when cells are damaged by a disease such as diabetes [11]. In peripheral diabetic neuropathy, the state of hyperglycemia controls apoptosis in many types of cells [12]. Also, apoptosis is enhanced and promoted via reactive oxidative species (ROS) generation due to hyperglycemia, which is a probable mechanism of glucose neurotoxicity. In addition, apoptosis is one of the potential mechanisms for hyperglycemia-induced neural cell death. The effects of apoptosis on neurons during diabetes has been assessed using in vitro and in vivo models. Apoptosis is seen as a strong factor that leads to damage of the nervous system [13]. Many cell types express the B-cell lymphoma 2 (Bcl2), which is anti-apoptosis gene [14]. Bcl-2 protects the cell against apoptosis with the notable exception of apoptosis induced by cytotoxic lymphocytes [15,16]. On the other hand, Bcl-2-like protein 4 (Bax) is a pro-apoptotic *Bcl-2*-family protein [17,18], which resides in the cytosol and translocates to the mitochondria upon induction of apoptosis [19]. Recently, Bax has been shown to induce cytochrome *C* release and caspase activation in vivo specifically in the CNS. Several studies showed that *Bcl-2* and *Bax’s* genes strongly link diabetes, CNS, and neuropathy.

There is a strong link between the cyclooxygenase (COX) enzyme, and apoptosis [20]. Proinflammatory stimuli induce the expression of COX-1 gene via growth factors. Also, its expression is induced via cytokines and mitogens in different cells [21]. The COX-1 expression level increases at the onset of diabetes and is associated with apoptosis. However, COX-1 levels decrease during the progress of diabetes. [22]. On the other hand, inflammation and carcinogenesis were associated with increased expression of COX-2. Protection of hepatocytes from several pro-apoptotic stimuli was mediated by the constitutive expression of COX-2. Several studies show that COX-2 expression protects against hyperglycemia-induced apoptosis [23].

The third endogenous gaseous transmitter, which was discovered after nitric oxide (NO) and carbon monoxide (CO), is hydrogen sulfide (H_2_S). It has been shown to exert several physiological effects on the body. Moreover, it has been linked to several pathological conditions [24]. Although H_2_S was viewed as a toxic environmentally hazardous gas, a lot of research has shown that mammalian tissues synthesize H_2_S, serving various key regulatory functions [25,26,27]. The benefits of anti-inflammatory, antioxidant, and cytoprotective functions of H_2_S were shown in the neuronal, cardiovascular, respiratory, and reproductive systems [28]. The pathogenesis of β-cell dysfunction results from changes in H_2_S stability, which occurs in response to T1DM and T2DM [25,26]. H_2_S generation was reduced in the early stage of diabetes in rats, and the decrease in H_2_S production was associated with an increase in ROS and pathogenesis of renal dysfunction [29]. Several lines of evidence suggest that elevated extracellular glucose leads to changing H_2_S levels, which contributes to the pathogenesis of endothelial cells by changing their organization pattern via increasing vascular endothelial growth factor [30]. More importantly, for the scope of this study, exogenous administration of H_2_S has been shown to exert antiallodynic and antihyperalgesic effects in diabetic rats [4,7]. 

Taking into consideration the above information from literature, we hypothesize that H_2_S has neuroprotective effects against T1DM-induced apoptosis and peripheral diabetic neuropathy in rats. Hence, the proposed study’s objective was to understand the mechanism by which H_2_S modulates apoptosis and diabetic neuropathy. Therefore, we investigated the effects of GYY4137, which is a water-soluble slow-releasing H_2_S donor that has potential vasodilation and anti-inflammatory properties on the expression of antiapoptotic and pro-apoptotic genes, namely *Bcl2* and *Bax*, the expression of antiapoptotic and pro-apoptotic proteins, such as caspases 3, 7 and 9, apoptosis, monocytes, macrophages, and endothelial cells, COX-1 and COX-2 expression in the spinal cord tissue of diabetic rats and the association of sensory neuron apoptosis and neurobehavioral defects such as hypoalgesia in a model of STZ-induced peripheral diabetic neuropathy.

## 2. Results

The glucose level of all animal groups did not show any significant difference during week 1. However, all diabetic rats, GYY4137-treated and non-treated, were hyperglycemic at week 2 and at the end of the study period (week 4), shown by significantly elevated blood glucose levels (*p* = 0.0014). On the other hand, both control groups had normal glucose levels, and there was no significant difference between them (Figure 1). There were no significant differences in glucose levels upon GYY4137 treatment in both control and diabetic groups.

Data from Figure 2 show no significant difference in body weight between the four groups in week 1. In contrast, there was a significant increase in rats’ body weights of both non-treated control and GYY4137-treated control groups during week 4. There was a significant decrease in diabetic rats’ body weight compared to the control group (*p* < 0.001). On the other hand, the GYY4137-treated diabetic group maintained stable body weight with a significantly (*p* = 0.025) higher body weight compared to the non-treated diabetic group during week 4. Diabetes induced weight loss, while treatment with GYY4137 prevented weight loss.

### 2.1. Effect of Treatment of Streptozotocin-Diabetic Rats with GYY4137 on Neurobehavioral Sensory Tests

The results of the Von Frey test, paw pressure test and hot plate test were similar to our previous results [7]. Diabetes induced tactile allodynia, mechanical hyperalgesia and thermal hypoalgesia, which were attenuated by treatment with GYY4137 (see Appendix A).

#### Tail Flick

Diabetic animals developed thermal hypoalgesia i.e., there was a significant (*p* < 0.001) increase in tail flick latency to radiant heat in the tail flick test in diabetic rats compared to control rats. Treatment with GYY4137 prevented the development of thermal hypoalgesia i.e., there was no significant difference between the GYY4137-treated diabetic group and the control group (*p* > 0.05). In addition, the tail flick latency to radiant heat of the GYY4137-treated diabetic group was significantly lower (*p*  <  0.001) in comparison to diabetic animals (Figure 3). 

### 2.2. Effect of Treatment of Streptozotocin-Diabetic Rats with GYY4137 on Transcripts and Proteins of Various Cell Markers and Molecules

#### 2.2.1. CD31 Expression

The staining of CD31, endothelial cells specific expression marker, was weak in the control groups (Figure 4). The immunostaining for CD31 was significantly increased in the spinal cord following the injection of STZ (*p* = 0.014, Figure 4A). The number of CD31-positive cells in the spinal cord was significantly decreased in the GYY4137-treated diabetic group compared with the non-treated diabetic group (*p* = 0.037, Figure 4B).

#### 2.2.2. CD68 Expression

The immunostaining for CD68, a marker for monocytes, macrophages, and some endothelial cells, was weak in the control groups (Figure 5A). Whereas the staining was robust, and the number of CD68-positive cells significantly increased in the spinal cord following the injection of STZ (*p* = 0.02, Figure 5A,B). The number of CD68-positive cells in the spinal cord was significantly decreased in the GYY4137-treated diabetic group compared with the non-treated diabetic group (*p* = 0.034, Figure 5B).

#### 2.2.3. Bcl2 Expression

##### Effect of Treating STZ-Diabetic Rats with GYY4137 on B-Cell Lymphoma 2 (Bcl2) Gene and Protein Expressions

The relative mRNA expression of the anti-apoptosis gene *Bcl2* was significantly suppressed in the diabetic group compared with the control group (*p =* 0.037). It was significantly restored/increased in the GYY4137-treated diabetic group compared with the non-treated diabetic group (*p* = 0.028, Figure 6A). The immunostaining for Bcl2 was robust in the spinal cord of control groups (Figure 6B). In contrast, the staining was weak in the diabetic group and significantly lower than in the control group (*p* < 0.001). The number of Bcl2-positive neurons in the spinal cord was significantly increased in the GYY4137-treated diabetic group compared with the non-treated diabetic group (*p* = 0.042, Figure 6C).

#### 2.2.4. Bax Expression

##### Effect of Treating STZ-Diabetic Rats with GYY4137 on Bcl-2-like Protein 4 (Bax) Gene and Protein Expressions

The relative mRNA expression of the pro-apoptosis gene *Bax* was significantly upregulated in the diabetic group compared with the control group (*p* = 0.012). It was significantly decreased in the GYY4137-treated diabetic group compared with the non-treated diabetic group (*p* = 0.0032, Figure 7). The immunostaining of Bax, a pro-apoptosis Bcl-2 family member, was significantly increased in the spinal cord of diabetic groups (*p* = 0.021, Figure 7B,C). In contrast, the staining was not significantly increased in the spinal cord of GYY4137-treated diabetic animals compared with control groups (*p* > 0.05, Figure 7B,C).

The ratio of Bcl2 to Bax expression represent a cell death switch, which determine the life of the cells in response to an apoptotic stimulus; an increased Bcl2/Bax ratio increase the cellular resistance to apoptotic stimuli, leading to decreased cell death. While, a reduced Bcl2/Bax ratio, which was decreased in diabetic group, leads to increased cell death (Figure 8). 

### 2.3. Effect of Treatment of Streptozotocin-Diabetic Rats with GYY4137 on Protein Expression of COX-1, COX-2, Caspase-7 and Caspase-9

#### 2.3.1. COX-1 Immunoblotting

The band size of COX-1 is ~70 kDa, as seen in Figure 9A. The densitometry analysis of the expression of the COX-1 protein showed that it was significantly decreased (*p* < 0.001) in diabetic animals compared to control animals. The expression of the COX-1 protein was significantly (*p* = 0.015) higher in the GYY4137-treated group compared to the non-treated diabetic group (Figure 9B).

#### 2.3.2. COX-2 Immunoblotting

The band size of COX-2 is ~72 kDa, as seen in Figure 10A. The densitometry analysis of the expression of the COX-2 protein showed that it was significantly increased (*p* < 0.013) in diabetic animals compared to control animals. On the other hand, the expression of the COX-2 protein was significantly (*p* < 0.002) lower in the GYY4137-treated group compared to the non-treated diabetic group (Figure 10B). In addition, there was no significant difference between the COX-2 protein concentration of the control group and that of the GYY4137-treated diabetic group. 

#### 2.3.3. Caspase-7 Immunoblotting

The band size of caspase-7 is ~33 kDa, as seen in Figure 11A. Densitometry analysis showed a significantly increased amount of caspase-7 protein in the diabetic group compared to control groups (*p* < 0.012). Conversely, the expression of caspase-7 protein was significantly lower in the diabetic GYY4137-treated group compared to the non-treated diabetic group (*p* < 0.017). Also, the data demonstrate no discernible difference between the control groups’ and the diabetic GYY4137-treated groups’ levels of caspase-7 protein Figure 11B.

#### 2.3.4. Caspase-9 Immunoblotting

The band size of caspase-9 is ~40 kDa as seen in Figure 12A. Densitometry analysis showed a significant increase amount of caspase-9 protein in the diabetic group compared to the control groups (*p* < 0.031). Conversely, the expression of caspase-9 protein was significantly lower in the diabetic GYY4137-treated group compared to the non-treated diabetic group (*p* < 0.042). Also, the data of caspase-9 protein concentration demonstrate no discernible difference between the control groups and the diabetic GYY4137-treated group Figure 12B.

## 3. Discussion

Research focusing on H_2_S has advanced significantly and quickly [31]. In particular, the antioxidative, anti-inflammatory, and antiapoptotic characteristics of H_2_S have drawn the attention of various researchers [32,33,34,35]. H_2_S donors have been recognized for their protective effects against diseases such as cancer, hypertension, myocardial ischemia, osteoarthritis, gastrointestinal disorders, hyperglycemia-induced injury to microvascular endothelial cells [36], osteoporosis [37], and osteoarthritis [38]. Such donors have also been reported to suppress diabetes-accelerated atherosclerosis [39]. Furthermore, they have proven cytoprotective and anti-inflammatory abilities. By reducing inflammation, oxidative stress, and apoptosis, H_2_S delays the progression of diabetic retinopathy, diabetic nephropathy, and diabetic cardiomyopathy [40].

This study highlights the effects of treatment with the slow-releasing H_2_S donor GYY4137 on diabetic neuropathy and the mechanism by which H_2_S modulates apoptosis and diabetic neuropathy associated behavioral changes. 

### 3.1. The Outcomes of H_2_S Treatment on Rats with Diabetes Induced by Streptozotocin with Regards to Body Weight, and Blood Glucose Level

In rodent DNP models, STZ is frequently used to induce type 1 diabetes. Streptozotocin-induced diabetic rats are frequently used as an animal model to explore the mechanism of type 1 DNP’s early phases. In this study, following STZ injection, diabetic rats had hyperglycemia and reduced body weight. Although H_2_S had been proven to have cytoprotective and anti-inflammatory abilities [7], it did not affect hyperglycemia but protected against body weight loss. 

GYY4137 has been shown to prevent lipolysis in mice without increasing fat mass [41]. Furthermore, GYY4137 was reported to protect H9c2 cells, which are a cell model used as an alternative for cardiomyocytes, from cytotoxicity caused by elevated glucose levels [42]. The current study’s findings are in line with these previous studies, demonstrating that diabetic rats treated with H_2_S maintained their basal weight compared to the non-treated diabetic group, which lost weight. The glucose-level data obtained in the current study show that H_2_S did not cure hyperglycemia since the GYY4137-treated diabetic rats’ glucose levels were notably high, similar to non-treated diabetic rats. Moreover, Szabo and colleagues have reported that the pancreas’ beta cells, which secrete insulin, are susceptible to injury as a side effect of H_2_S [43]. Thus, GYY4137 has no positive effects on the pancreas but protects against diabetes/hyperglycemia-induced weight loss.

### 3.2. Antiallodynic/Anti-Hyposensitivity Effect of the H_2_S Donor on STZ-Induced Diabetic Rats

In this study, STZ-induced diabetic rats had mechanical allodynia and hyperalgesia, and thermal hyposensitivity, which were ameliorated by treatment with GYY4137 similar to our previous findings [7]. Mechanical allodynia and hyperalgesia, and thermal hyposensitivity are symptoms of DNP observed in rodent STZ-induced DM models [7,44,45], which were reproduced in this study. In this study, pretreatment with GYY4137 considerably protected against mechanical allodynia and hyperalgesia, and thermal hyposensitivity caused by STZ. Behavioral changes began the second week after rats were given STZ to induce diabetes and persisted throughout the experimental period. These findings are consistent with earlier studies [46].

Sensory abnormalities such as unpleasant aberrant sensations (dysesthesia), pain in response to a stimulus that does not typically cause pain (allodynia), and an increased sensitivity to feeling pain and an extreme response to pain (hyperalgesia) are characteristics of neuropathic pain [47]. In the current study, diabetic rats showed a reduction in thermal nociception perception, which may be related to nerve damage caused by the onset of DN [48]. Overall, the findings of this study imply that H_2_S reduced the symptoms of painful neuropathy associated with diabetic neuropathy by preventing the onset of mechanical allodynia and hyperalgesia, and thermal hyposensitivity, which are the defining symptoms of diabetic neuropathic pain.

There is inconsistency with regards to the effects exogenous of administration of H_2_S to rodents with STZ-induced diabetes. Two studies from the same group showed that intraplantar or subcutaneous injection on the paw of a fast-releasing H_2_S donor, NaHS, caused hyperalgesia in formalin-induced nociception in diabetic rats and tactile allodynia in diabetic rats via activation of TRPV1, TRPA1 and TRPC channels, and subsequent loss of intraepidermal fibers [49,50]. On the other hand, one study showed that inhalation of H2S gas attenuated the diabetes-induced mechanical allodynia and thermal hyperalgesia through NO/cGMP/PKG pathway and µ-opioid receptor [4]. Another study showed that intraperitoneal injection of a slow-releasing H_2_S donor, GYY4137, prevented the development of mechanical allodynia and hyperalgesia in diabetic rats possibly through inhibition of glial cell activation [7]. The main difference between the studies that showed hyperalgesic effects of H_2_S administration and those that showed antiallodynic and antihyperalgesic effects of H_2_S administration in diabetic rodents is that the former group administered the H_2_S donor locally in the paw while the latter administered the drugs systemically. Thus, it is plausible that local administration produced hyperalgesia through activation of TRP channels while systemic administration produced antiallodynic and antihyperalgesic effects through various mechanisms including NO/cGMP/PKG pathway, µ-opioid receptor, and inhibition of glial cells. A good example of an endogenous molecule that can have such a dual effect depending on where it is released is serotonin, which in the periphery is inflammatory and hyperalgesic while in the CNS is antinociceptive or antihyperalgesic. It is also possible that the dual effects of H_2_S, i.e., antinociceptive and hyperalgesic, could be due to the dose at the site of action. A recent study showed that dopamine has dual effects in the periphery, where lower doses are antinociceptive while higher doses are hyperalgesic due to the activation of different dopamine receptors [51]. Further research is necessary to explore the possible reason for the dual effect of H_2_S on nociception.

### 3.3. Effect of H_2_S on the Levels of Endothelial, and Monocyte/Macrophage Cells Markers in the Spinal Cord

Infiltration of macrophages and monocytes into the spinal cord plays a vital role in the pathogenesis of DNP. Our study shows that H_2_S reduces the infiltration of monocyte and macrophage cells into the spinal cord. Streptozotocin injections induced DN and increased the infiltration of monocytes/macrophages into the spinal cord, compared to non-diabetic mice, as evidenced by the increase of their marker CD68. This finding is consistent with a previous study [52], which reported that macrophage cells infiltrated the spinal cord as a result of diabetes. Moreover, another group [53] showed that the activation and infiltration of macrophage cells were related to mechanical allodynia in STZ-treated rats. The current study showed that monocyte/macrophage cells were increased in the spinal cord and correlated to neurobehavioral test results. In dyslipidemia, H_2_S has been proven to reduce of infiltration process of monocyte/macrophage cells on the vascular surface [54]. In this study, CD31 expression was increased, which means that endothelial cells in histological tissue sections of the spinal cord were increased. This is in agreement with Chen-Yuan Gong and his colleagues [55] who found that CD31 expression was increased in the retinas of diabetic rats. On the other hand, Ved and colleagues [56] found that CD31 was downregulated in the spinal cord of the diabetic group. On the other hand, GYY4137 decreased the expression of CD31, possibly indicating a decrease in the inflammation process. However, the infiltration process of macrophages, monocytes and endothelial cells in the spinal cord needs to be further elucidated.

### 3.4. Effect of H_2_S on Cyclooxygenase-1 and -2 Enzymes

A constitutive enzyme, COX-1, mediates physiological processes, while the COX-2 enzyme is induced by inflammatory stimuli. COX-1 provides prostaglandins, which are needed for homeostatic functions such as gastric cytoprotection and hemostasis. Also, it has a vital role in controlling pain, fever, tumorigenesis, and inflammation [57]. Whereas reproductive tissue, the brain, and the kidney express COX-2, which plays a vital role in renal function. It also works as a protective in GI, in which it is induced rapidly in case of inflammation and ulcers [58]. Persaud and colleagues [59] found that COX-2 expression in human islets of Langerhans was upregulated as a result of diabetes. Fang and colleagues [60] observed that the expression of constitutive COX-1 was decreased in the sciatic nerves of rats with STZ-induced diabetes. The results of this study, which show decrease in expression of COX-1 in the spinal cord of the diabetic group, support these two studies on this topic. In addition, our results support a previous study (Fang, 1997), which found that levels of COX-1 in diabetic rat sciatic nerve and thoracic aorta were reduced significantly by 39% and 55%, respectively and another previous study [61], which found that elevation of spinal COX-2 expression is associated with hyperalgesia in diabetic rats. 

Several disease models have found H_2_S to be cytoprotective. Burguera and colleagues showed that H_2_S has a vital role in decreasing COX-2 enzyme expression in human articular chondrocytes [62]. This result is supported by our results, which showed that expression of the COX-2 protein was significantly decreased in the GYY4137-treated group compared to the non-treated diabetic group. Our results also support the findings of Wallace et al. [63], which showed that H_2_S has a role in decreasing COX-2 enzyme expression in the gastrointestinal tract. Regarding the COX-1 enzyme, our results indicate that H_2_S had a vital role in the increased expression of the COX-1 enzyme in the spinal cord of the GYY4137-treated diabetic group. To our knowledge, no previous study reported that H_2_S has a role in COX-1 enzyme expression. 

### 3.5. Antiapoptotic Effect of H_2_S on Rats with STZ-Induced Diabetes

Yonguc and colleagues showed that diabetes leads to the stimulation of apoptosis in the hippocampi of rats with STZ-induced diabetes. They reported that the expression of *Bcl2* and *Bcl2l1* (Bcl-XL) genes was significantly decreased in the hippocampi of diabetic rats, whereas the expression of *Bax*, *caspase-3*, *-9*, and *-8* genes was significantly increased compared to control rats [64]. Sun et al. [35] showed that hippocampal neurons degenerated in the hippocampi of rats with STZ-induced diabetes. Later, we [7] confirmed that spinal cord sensory neurons were downregulated in diabetic rats compared to control rats. Our results align with and support these three studies. Bcl2 gene and protein expressions were decreased in the diabetic group compared to the control group. In contrast, the Bax gene and protein expressions were increased in the diabetic group compared to the control group. The protein levels of caspases 7 and 9 were highly expressed in the diabetic group compared to the control group. All of these suggest increased apoptosis in the spinal cord of diabetic rats.

Polhemus and Li showed that H_2_S plays a novel role in the apoptosis process [65]. They demonstrated robust molecular crosstalk and signaling between H_2_S and NO. The authors also found that heart failure patients are deficient in H_2_S and NO, two important molecules for cardiovascular homeostasis. In their study, they used the H_2_S prodrug SG1002, which releases H_2_S and attenuated the apoptosis process [65]. In another experiment, George, A.K and his colleagues [66] proved that H_2_S is an effective antioxidant agent and can help treat aging eyes by alleviating stress and inflammation. Stress and inflammation may lead to apoptosis. Our results are aligned with these two studies. *Bcl2* gene expression was relatively increased in the GYY4137-treated diabetic group compared to the non-treated diabetic group. At the same time, *Bax* gene expression was relatively decreased in the GYY4137-treated diabetic group compared to the non-treated diabetic group. In addition, the protein expression levels of caspases 7 and 9 were significantly decreased in the GYY4137-treated diabetic group compared to the non-treated diabetic group. However, the expression of *Bax* and *Bcl2* genes did not significantly differ between the GYY4137-treated diabetic group and the control group. Likewise, the protein expression levels of caspases 3 and 7 did not significantly differ between the GYY4137-treated diabetic group and the control group. This agrees with previous studies [67], which found that H_2_S enhances cell proliferation and antiapoptosis in PLC/PRF/5 hepatoma cells. They found that the down-expression of caspase-3 increased cell viability and decreased the number of apoptotic cells [67].

### 3.6. Summary

In summary, activation of apoptosis in response to inflammation is a hallmark of neurological disorders. Under diabetic conditions, the reactive oxidative species leads to DNA damage, which then activates the p53 gene. This activation will lead to the activation of *Bax* gene, which stimulates the mitochondria, and inhibition of *Bcl2* gene. The inhibition of *Bcl2* gene expression also increases the activation and expression of *Bax* gene. The expression and stimulation of Bax will activate the mitochondrial membrane that will secrete cytochrome C. This secretion with Apaf1 results in activation of pro-caspase-9 into caspase-9. Then, this activation will activate pro-caspase-3 into caspase-3. After that, the caspase-3 activates the nuclease enzyme leading to apoptosis. Circulating levels of H_2_S are depressed in diabetic rats, as previously mentioned. H_2_S shows a significant inhibitory effect of spinal apoptosis process activation, which consequently upregulates the expression of antiapoptotic genes and proteins. Also, H_2_S attenuates the development of STZ-induced diabetes-related neuropathic behaviors such as allodynia, hyperalgesia, and hyposensitivity. It is important to investigate more how spinal apoptosis and infiltration of macrophages, monocytes, and endothelial cells were deactivated following H_2_S treatment and the effect of H_2_S on other enzymes such as COX-1 and COX-2 in the spinal cord. Answers to these questions could lead to new therapeutic strategies to overcome diabetic neuropathy in patients.

## 4. Methods and Materials

The methods used in the experiments are shown in a schematic representation (Figure 13).

### 4.1. Animals

Sprague Dawley male rats (2–3 months old; ~300–400 g) were ordered from the Animal Resources Center of the Health Sciences Center of Kuwait University. They were kept three rats per cage in plastic cages with hardwood chips for bedding, which have a high absorbance rate to prevent the accumulation of waste. The rats had access to pelleted chow and water *ad libitum*. The light/dark cycle was maintained for 12 h for all of the rats. The hours of this cycle were 7 a.m. to 7 p.m. The temperature was maintained at 25 ± 1 °C. Finally, the rats were separated and grouped accordingly (Table 1).

### 4.2. Drug and Chemicals

GYY4137 is a water-soluble slow-releasing H_2_S donor that was originally synthesized and described by Li et al. [68]. It has potential vasodilation and anti-inflammatory properties [7]. GYY4137 was synthesized and characterized as previously described [69]. In brief, GYY4137 was synthesized as follows: Dropwise addition of morpholine (40 mmol) to a Lawesson’s reagent solution (8.0 mmol) in anhydrous dichloromethane (DCM, 12 mL) was performed under nitrogen gas at room temperature. The reaction mixture was then stirred for two hours. A pure white solid product (66% yield; melting point 156–159 °C) was produced after the precipitate was dried, washed several times with anhydrous DCM, and collected by suction filtration. After synthesis, proton nuclear magnetic resonance spectrometry was carried out at the General Facilities Science (GFS), College of Science, Kuwait University, Kuwait, to confirm the product’s structure and purity. All chemicals were purchased from Sigma Aldrich, St. Louis, MO, USA.

### 4.3. Induction of Diabetes and H_2_S Donor Administration

The rats were fasted overnight for 12 h. before a single dose of STZ (Sigma-Aldrich, Taufkirchen, Germany), freshly dissolved in a citrate buffer solution with a pH of 4.5, was injected intraperitoneally (i.p.) at a dose of 55 mg/kg to induce diabetes. Animals had unrestricted access to food and water following STZ administration. The glucose level was checked 48 h. after STZ injection using a glucometer to estimate the blood sugar levels (BSL). The animals with a BSL of >300 mg/dL were considered diabetic and used further for the study. The H_2_S donor GYY4137 was administered by intraperitoneal injection once daily for 28 consecutive days. Control groups received the same volume of physiological saline (vehicle for GYY4137) instead of drug once daily over the same period. The dose of GYY4137, 50 mg/kg, was selected based on previously published data [7,69]. The Kuwait University Health Science Center Animal Research Ethics Committee approved the animal experimental procedures used in this study (Ref: 23/VDR/EC/Date: 3 April 2021).

### 4.4. Neurobehavioral Sensory Tests

The Von Frey test, paw pressure test, and hot plate test were carried out as previously described Shayea in 2020, [7]. These neurobehavioral tests are in Appendix A [7,70,71,72,73,74,75,76,77,78,79].

#### Tail-Flick Test

To assess the spinal response, a tail-flick test was conducted using Analgesia Meter Apparatus (Ugo-Basile, Gemonio, Italy) [80] at 14 and 28 days after STZ administration. The tail-flick test was performed by placing a rat on a platform. This platform contains a heat-generating beam of light. The animal was gently restrained by being wrapped with a towel. After that, the rat was positioned where the tail was exposed to a narrow beam of light. In a brief period, when the heat that was produced by the beam of light, the animal becomes uncomfortable and reflexively moves its tail away from the heat source. The photocell was blocked when the animal’s tail was in the test position. When the animal moved its tail, the photocell was activated and turned off the timer and energy source. The time between exposure to the heat source and the tail flicking was recorded as the dependent variable. The beam’s intensity was adjusted to produce a reaction latency of approximately 3–6 s. The test was generally performed three times for each animal, at 3- to 5-min intervals, and the average of the three readings was recorded. The light beam was delivered approximately 50 mm from the tip of the tail for rats [81].

### 4.5. Perfusion and Tissue Processing

The animals were sacrificed on the 28th day of the experiment. Briefly, chloral hydrate (Sigma Aldrich, St. Louis, MO, USA; 400 mg/kg) was administered i.p. to anesthetize the animals. The animals were then transferred to a ventilated bench where the thorax was opened to expose the heart and a cannula was inserted into the left ventricle. A cut was also carried out in the upper anterior region of the right atrium at the same time for draining blood. Next, the blood was rinsed from the body by introducing normal saline and allowing it to circulate. Then, the tissue fixation process was started by introducing freshly prepared 4% formaldehyde (Sigma Aldrich, St. Louis, MO, USA) in 0.1 phosphate buffer (pH = 7.4). The bottles containing the perfusion solutions were suspended roughly 1.6 m above the animal to administer the solutions under pressure. After the fixation step, the brain and spinal cord’s relevant parts (L4, L5 and L6) were dissected out. The first cervical roots (C1) of the spinal segments were used as a landmark to determine L4, L5 and L6 segments. After removal, the brains and spinal cords were post-fixed overnight in 4% paraformaldehyde.

### 4.6. RNA Isolation and Reverse Transcription (RT)

To purify total RNA from frozen tissue, TRIzol (Invitrogen, Carlsbad, CA, USA) was used. This was carried out following the manufacturer’s instructions. Total RNA concentration was measured spectrophotometrically at 230, 260, and 280 nm using Nanodrop spectrophotometer. The samples were kept at −80 °C. Following the manufacturer’s instructions, the high-capacity cDNA reverse transcription kit was used to create cDNA from 2 μg of pure RNA (Thermo Fisher Scientific, Waltham, MA, USA).

### 4.7. Real-Time PCR

TaqMan^®^ gene expression assays (Thermo Fisher Scientific, Waltham, MA, USA) that had been validated were used for real-time PCR reactions. This technique was used to determine the relative gene for B-cell lymphoma 2 and Bcl-2 associated X. As an endogenous control, β-actin was used. TaqMan universal master mix (Thermo Fisher Scientific, Waltham, MA, USA), cDNA template, TaqMan assay, and nuclease-free water are the materials of the real-time PCR reaction. They were used to complete the 20 μL final volume. These were put in a 96-well reaction plate. After that, the plate was placed in the 7500 Sequence Detection System (Applied biosystems, Waltham, MA, USA). The cycling parameters recommended by the manufacturer were used. The gene expression of β-actin in the experimental groups was used as a calibrator. The relative mRNA expression was measured and calculated using the 2^−CT^ equation. The sequences of the primers used are shown in Table 2. 

### 4.8. Western Blot

The L4, L5, and L6 segments of the spinal cord were quickly isolated. They were placed directly in liquid nitrogen. After that, these segments were stored at −80 °C. The frozen segments were mixed with radioimmunoprecipitation assay (RIPA) buffer (Santa Cruz Biotechnology, Dallas, TX, USA). Then, per mL of 1× RIPA lysis buffer, 10 μL of the PMSF solution, 10 μL of the sodium orthovanadate solution, and 10–20 μL of the protease inhibitor cocktail solution were combined. Three mL of RIPA was required for one gram of tissue. The homogenized sample’s total protein was determined using the BioTek protein estimation instrument. Four to twenty % Mini-PROTEAN TGX Precast Protein Gels were used for running the western blot. Bio-Rad precision plus protein kaleidoscope (Bio-Rad Laboratories, Cat No. 1610375, Dubai, United Arab Emirates), was used as the ladder. 50 μg of the sample was loaded into each well and ran under these parameters, Voltage (100 V), 50–75 min. The PVDF membrane (Bio-Rad Laboratories, Cat. No. 162-0177) was soaked in methanol (10 min), distilled water (10 min), and 1× transfer buffer (#NP00061, Invitrogen, Thermofisher, Waltham, MA, USA) (10 min). The gel was transferred to the membrane at 75 V (2 gels) for 75 min of blotting. Next, the membrane was blocked in a blocking solution, which was a milk solution. This blocking was for 1 h at room temperature on a shaker. Western was used to determine the expression levels of various proteins such as Caspase 9 (sc-56079, mouse monoclonal—Santa Cruz Biotechnology, Dallas, TX, USA, Caspase 7 (sc-56063, mouse monoclonal—Santa Cruz Biotechnology, Dallas, TX, USA), COX-1 (sc-19998, mouse monoclonal—Santa Cruz Biotechnology, Dallas, TX, USA) and COX-2 (sc-376861, mouse monoclonal—Santa Cruz Biotechnology, Dallas, TX, USA). The film was placed on top of the membrane, which was maintained under the cassette in a Kapak pouch (Cat No. GE28-9068-35; Amersham, Bio-Rad Laboratories). 

### 4.9. Immunohistochemistry

Vascular endothelial cells were stained using the anti-CD31 antibody (Abcam, ab222783; Proteintech, 11265-1-AP with dilution 1:100). Monocytes/macrophages were stained by using anti-CD68 (Abcam, ab125212; Proteintech, 11265-1-AP with dilution 1:100). CD31 and anti-CD68 are also expressed in lymphocytes, fibroblasts, and endothelial cells. Also, to stain and quantify the number of neurons, the tissue sections were immunostained for nuclear-specific antigens (NeuN, Abcam, ab177487; Proteintech, 11265-1-AP with dilution 1:100) [7]. An Abcam kit (Cat # ab14933-100) was used for IHC in this experiment (PK-6200, Fremont, CA, USA). First, all samples were dewaxed through the three changes of xylene each for 5 min and then hydrated with alcohols until reaching 90% alcohol each for 3 min. Then, sections were washed in tap water, followed by 3% H_2_O_2_ (Fluka Chemika, Buchs, Switzerland) for 10 min at room temperature, followed by 5 min of washing in tap water. The slides were immersed in an antigen retrieval solution (citrate buffer pH 6.0) and cooked in a vegetable steam cooker for 20 min, then cooled for 20 min. The slides were washed in TBS for 5 min, followed by applying a blocking buffer containing 1% BSA (Cat. No. H0146; Sigma-Aldrich, St. Louis, MO, USA) for at least 20 min at room temperature. The excess blocking solution was removed, and the primary antibodies were applied according to the manufacturer recommendation; then slides were incubated overnight at 4 °C. Sections were washed in TBS for 5 min and covered with the secondary antibody for 30 min at room temperature. In between, samples were washed in TBS for five minutes. ABC complex (a biotinylated immunoglobulin) was applied on the slides for 30 min at room temperature, followed by washing for 5 min in the buffer. Finally, the slides were incubated in freshly prepared substrate DAB chromogen solution (DAB Kit, SK-4100, Vector Labs, Burlingame, CA, USA) for ten minutes in the dark, followed by washing in running water for five minutes. The tissue was put in hematoxylin counterstain, which is an additional dye for contracting the background for three minutes. Furthermore, samples were washed in running tap water for five minutes. In the end, samples were dehydrated through alcohol, cleared in xylene for three minutes each, then mounted in DPX and visualized through the Olympus microscope (Center Valley, PA, USA). 

### 4.10. Statistical Analysis

Data are shown as the mean ± standard error of the mean or standard deviation. Normality was tested by using the Shapiro-Wilk test. Data were analyzed using two-way analysis of variance (ANOVA) and one-way ANOVA with Bonferroni’s post hoc tests. This was carried out using Prism 5.0 (Graph Pad Software, Inc., La Jolla, CA, USA). The *p*-value < 0.05 was considered a statistically significant difference.

## 5. Conclusions and Recommendations

In summary, the current study demonstrated the potential protective properties of H_2_S, since a slow-releasing H_2_S donor prevented neuropathic pain behavior from developing in diabetic rats. The slow-releasing H_2_S donor also suppressed apoptosis activation pathways, attenuated infiltration/elevation of macrophage/monocyte and endothelial cell, and COX-2 enzymes, which were all linked to its protective effects in the spinal cord. H_2_S donor compounds may be used in the future to treat diabetic patients’ peripheral neuropathy. 

## Figures and Tables

**Figure 1 ijms-24-16650-f001:**
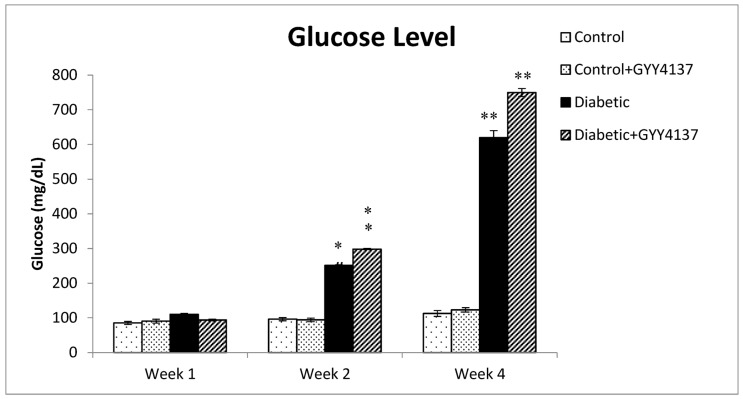
Blood glucose level of control, diabetic, and GYY4137-treated and non-treated groups. From week 1 to week 2, blood glucose levels in the diabetic and diabetic-treated groups were significantly ** p* = 0.031 higher than in the control groups, indicating hyperglycemia. Also, ** *p* = 0.0014 higher blood glucose levels of the diabetic treated group compared to control groups at week 4 (two-way ANOVA followed by Bonferroni post-test). Values are mean ± SD, *n* = 10–15.

**Figure 2 ijms-24-16650-f002:**
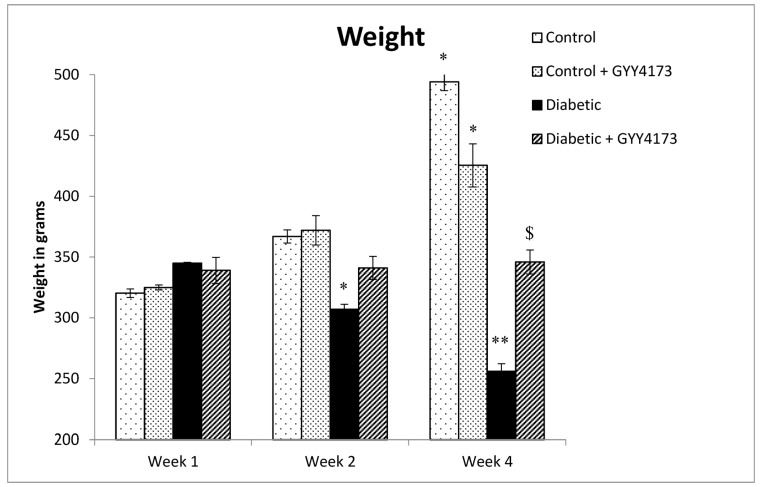
Body weights of the different experimental groups. Normal rats had increased body weights from week 1 to week 4. The diabetic group showed a significantly decreasing weight * *p* = 0.037 compared to control groups at week 2. The diabetic rats had weight loss ** *p* < 0.001. GYY4137-treated diabetic maintained stable weight with a significantly $ *p* = 0.025 higher body weight compared to the non-treated diabetic group during week 4 (two-way ANOVA followed by Bonferroni post-test). Values are mean ± SD, *n* = 10–15.

**Figure 3 ijms-24-16650-f003:**
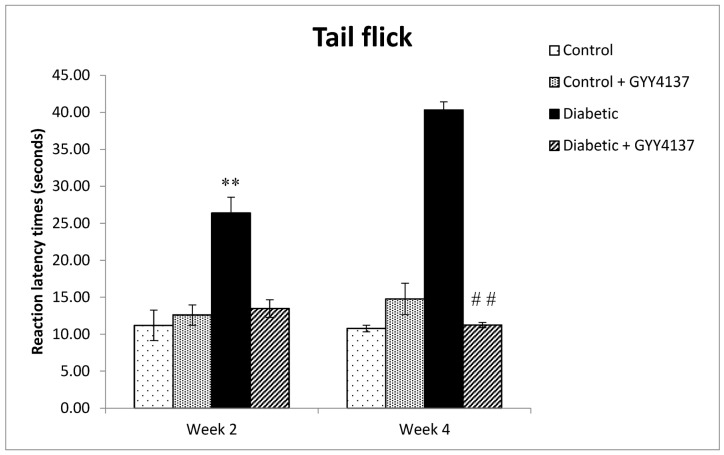
Response latency to radiant heat in the tail-flick test of the experimental groups at week 2 and week 4. (** *p* < 0.001) Diabetic group vs. Control group at week 4. At week 4 (## *p* < 0.001) Diabetic + GYY4137 group vs. Diabetic group (two-way ANOVA followed by Bonferroni post-test). Values are mean ± SD, *n* = 10.

**Figure 4 ijms-24-16650-f004:**
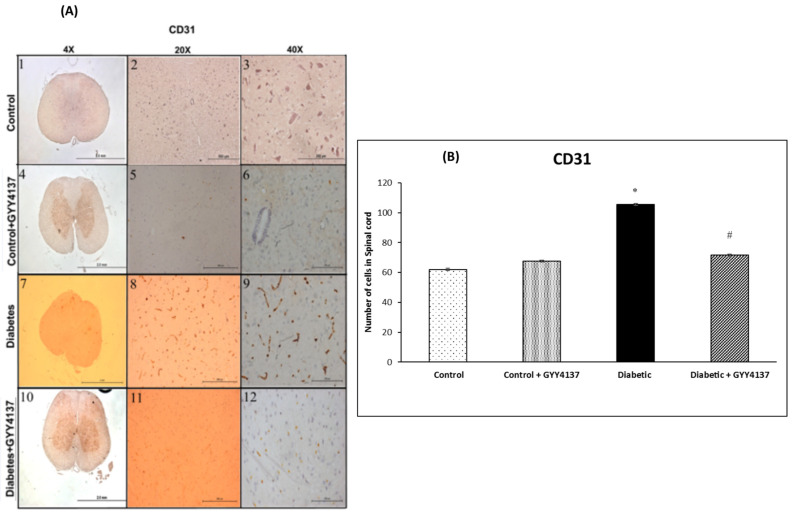
(**A**) Immunohistochemistry images showing the CD31-immunoreactive endothelial cells of the spinal cord in the different experimental groups at different magnifications (4×, 10× and 40×). The diabetic group showed more CD31-immunoreactive endothelial cells (panels 7–9) compared to the control group (panels 1–3). The GYY4137-treated control group (panels 4–6) were similar to the untreated control group. GYY4137-treated diabetic group (panels 10–12) showed fewer immunoreactive endothelial cells than the diabetic group. (**B**) The number of CD31-immunoreactive endothelial cells in the spinal cord of the experimental groups. The diabetic group showed a significantly higher number of immunoreactive endothelial cells compared to the control groups (* *p* = 0.014). The GYY4137-treated diabetic group showed a significantly (# *p* = 0.037) lower number of endothelial cells compared to the diabetic group (one-way ANOVA followed by Bonferroni post-test). Values are mean ± SD, *n* = 5.

**Figure 5 ijms-24-16650-f005:**
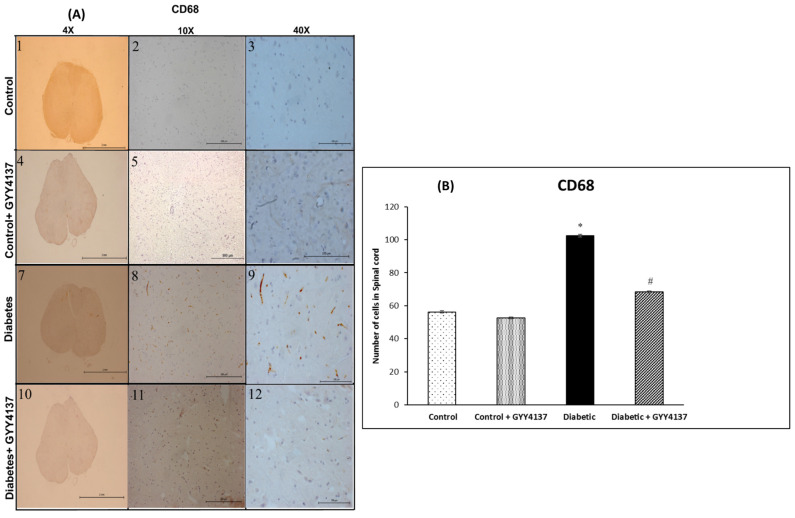
(**A**) Immunostaining images showing the CD68-immunoreactive monocytes/macrophage cells of the spinal cord in the different experimental groups at different magnifications (4×, 10× and 40×). The diabetic group (panels 7–9) showed a higher number of CD68-immunoreactive monocytes/macrophage cells compared to the control group (panels 1–3). The GYY4137-treated control group (panels 4–6) were similar to the untreated control group. The GYY4137-treated diabetic (panels 10–12) group showed fewer immunoreactive monocytes/macrophages compared to the diabetic group. (**B**) The number of CD68-immunoreactive monocytes/macrophage cells in the spinal cord of the experimental groups. The diabetic group showed a significantly higher number of immunoreactive monocyte/macrophage cells compared to control groups (* *p* = 0.02). Conversely, the GYY4137-treated diabetic group showed significantly # *p* = 0.034) fewer monocyte/macrophage cells than the diabetic group (one-way ANOVA followed by Bonferroni post-test). Values are mean ± SD, *n* = 5.

**Figure 6 ijms-24-16650-f006:**
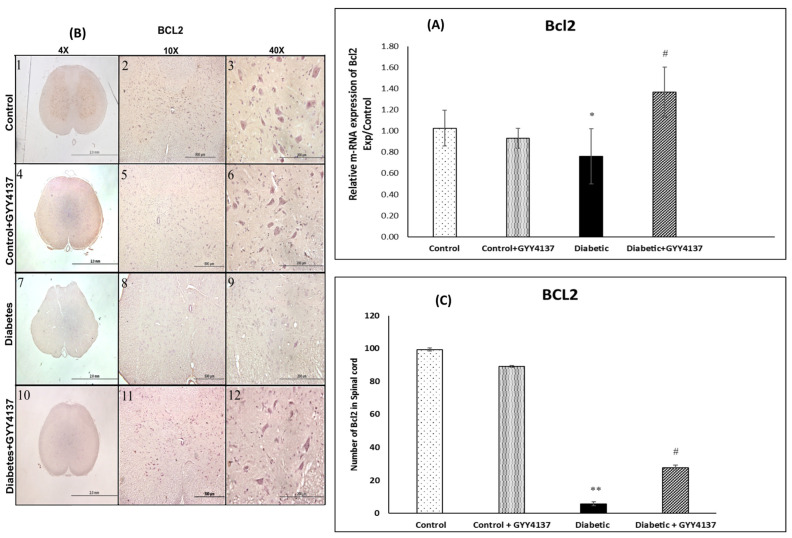
(**A**) The relative mRNA expression of the anti-apoptosis *Bcl2* gene was significantly suppressed in the diabetic group compared with the control group * *p* = 0.037. It was significantly restored/increased in the GYY4137-treated diabetic group compared with the non-treated diabetic group # *p* = 0.028. Values are mean ± SD, *n* = 5. (**B**) Photographs showing the Bcl2-immunoreactive neurons of the spinal cord in the different experimental groups at different magnifications (4×, 10× and 40×). Compared to the control group (panels 1–3), the diabetic group (panels 7–9) displayed a considerably decreased number of Bcl2-immunoreactive neurons. The GYY4137-treated control group (panels 4–6) were similar to the untreated control group. The GYY4137-treated diabetic group (panels 10–12) had higher number than the untreated diabetic group. (**C**) Graph comparing the number of Bcl2-immunoreactive neurons in the experimental groups’ spinal cords. Compared to the control groups, the diabetes group displayed a significantly decreased number of Bcl2-immunoreactive neurons (** *p* = 0.01). The GYY4137-treated diabetic group had significantly (# *p* = 0.042) higher number of neurons compared to the non-treated diabetic group. (one-way ANOVA followed by Bonferroni post-test). Values are mean ± SD, *n* = 5.

**Figure 7 ijms-24-16650-f007:**
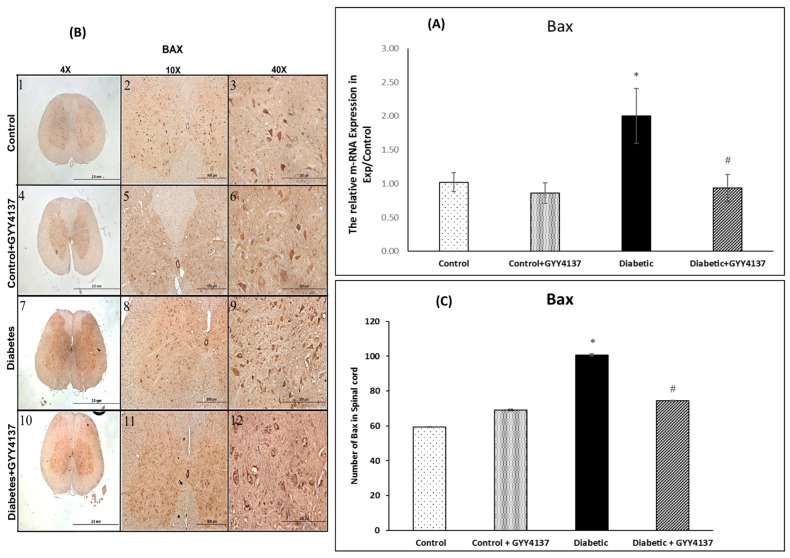
(**A**) The relative mRNA expression of the pro-apoptosis genes *Bax* was significantly increased in the diabetic group compared with the control group * *p =* 0.012. It was significantly decreased in the GYY4137-treated diabetic group compared with the non-treated diabetic group # *p =* 0.0032. (one-way ANOVA followed by Bonferroni post-test). Values are mean ± SD, *n* = 5. (**B**) Immunostaining images showing the Bax-immunoreactive neurons of the spinal cord in the different experimental groups at different magnifications (4×, 10× and 40×). The diabetic group (panels 7–9) showed more Bax-immunoreactive neurons than the control group (panels 1–3). On the other hand, the GYY4137-treated diabetic group (panels 10–12) showed no difference in the number of immunoreactive neurons compared to the GYY4137-treated control group (panels 4–6). (**C**) The number of Bax-immunoreactive neurons in the spinal cord of the experimental groups. The diabetic group had significantly more immunoreactive neurons than the control groups (* *p* = 0.021). GYY4137-treated diabetic group had significantly fewer neurons compared to the non-treated diabetic group (# *p* > 0.05) 028 (one-way ANOVA followed by Bonferroni post-test). Values are mean ± SD, *n* = 5.

**Figure 8 ijms-24-16650-f008:**
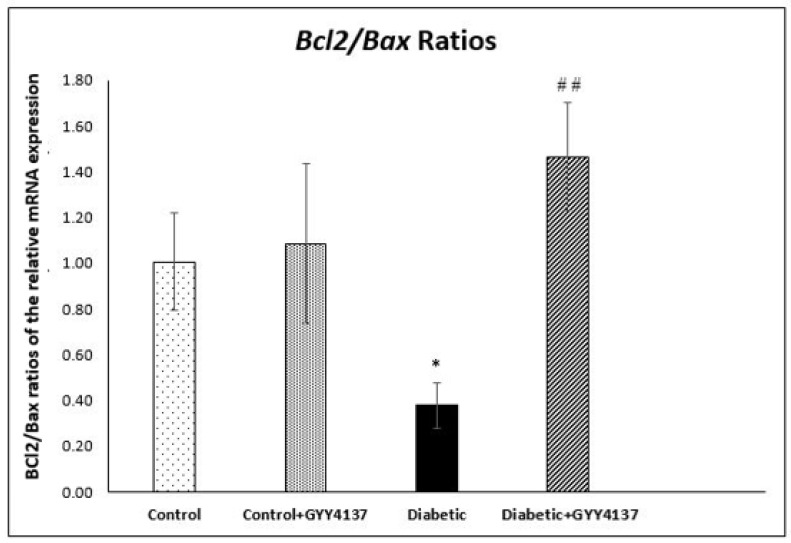
The ratios of the relative mRNA expression of the apoptosis related genes *Bcl2* and *Bax* genes were significantly suppressed in the diabetic group compared with the control group * *p* = 0.017. The GYY4137-treated diabetic group had significantly higher Bcl2/Bax ratios compared with the non-treated diabetic group ## *p* < 0.01 (one-way ANOVA followed by Bonferroni post-test). Values are mean ± SD, *n* = 5.

**Figure 9 ijms-24-16650-f009:**
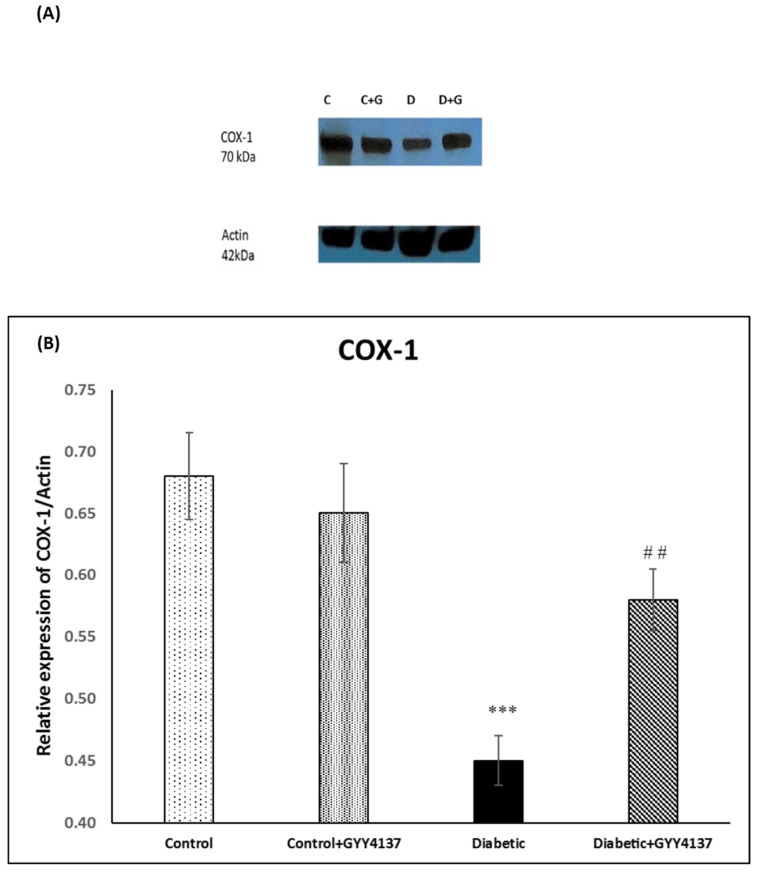
(**A**): Immunoblot of COX-1 in the lumbar region of the spinal cord. (**B**): Densitometry for COX-1 protein band expression. *** *p* < 0.001; diabetic group vs. control groups. ## *p* = 0.025; GYY4137-treated diabetic group vs. diabetic group. (one-way ANOVA followed by Bonferroni post-test). Values are mean ± SD, *n* = 5. C: control, D: diabetic, G: GYY4137.

**Figure 10 ijms-24-16650-f010:**
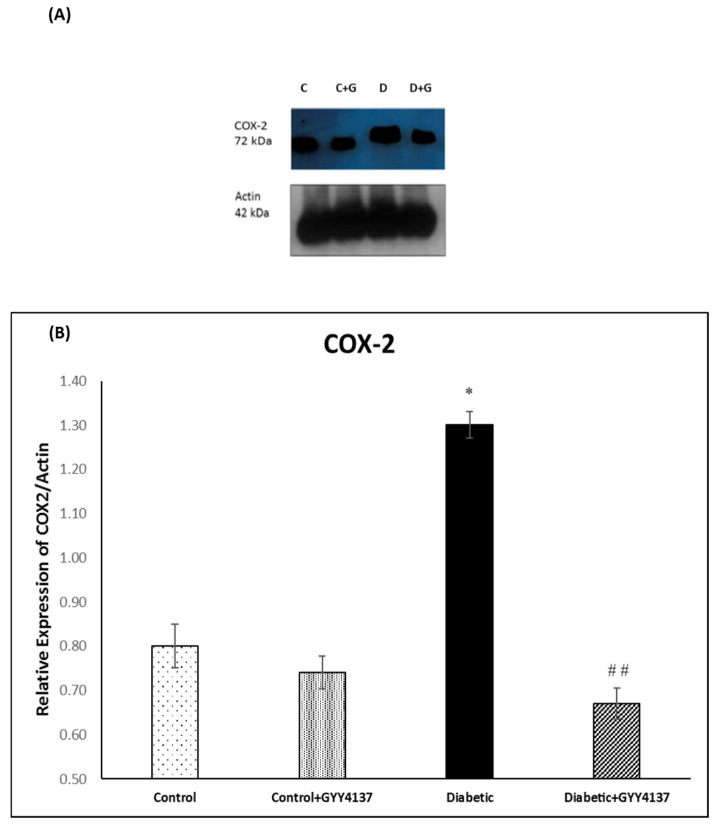
(**A**): Immunoblot of COX-2 in the lumbar region of the spinal cord. (**B**): Densitometry for COX-2 protein band expression. * *p* = 0.013; diabetic group vs. control groups. ## *p* = 0.002; GYY4137-treated diabetic group vs. diabetic group. (one-way ANOVA followed by Bonferroni post-test). Values are mean ± SD, *n* = 5. C: control, D: diabetic, G: GYY4137.

**Figure 11 ijms-24-16650-f011:**
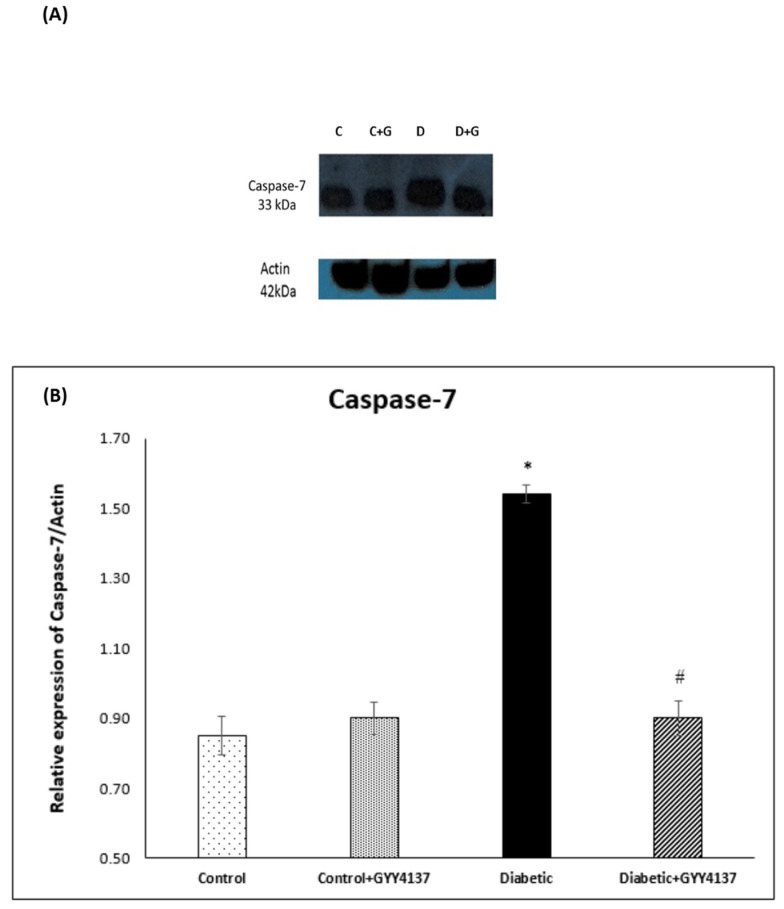
(**A**): Immunoblot of Caspase-7 in the lumbar region of the spinal cord (**B**): Densitometry for Caspase-7 protein band expression. * *p* = 0.012; diabetic group vs. control groups. # *p* = 0.017; GYY4137-treated diabetic group vs. diabetic group. (one-way ANOVA followed by Bonferroni post-test). Values are mean ± SD, *n* = 5. C: control, D: diabetic, G: GYY4137.

**Figure 12 ijms-24-16650-f012:**
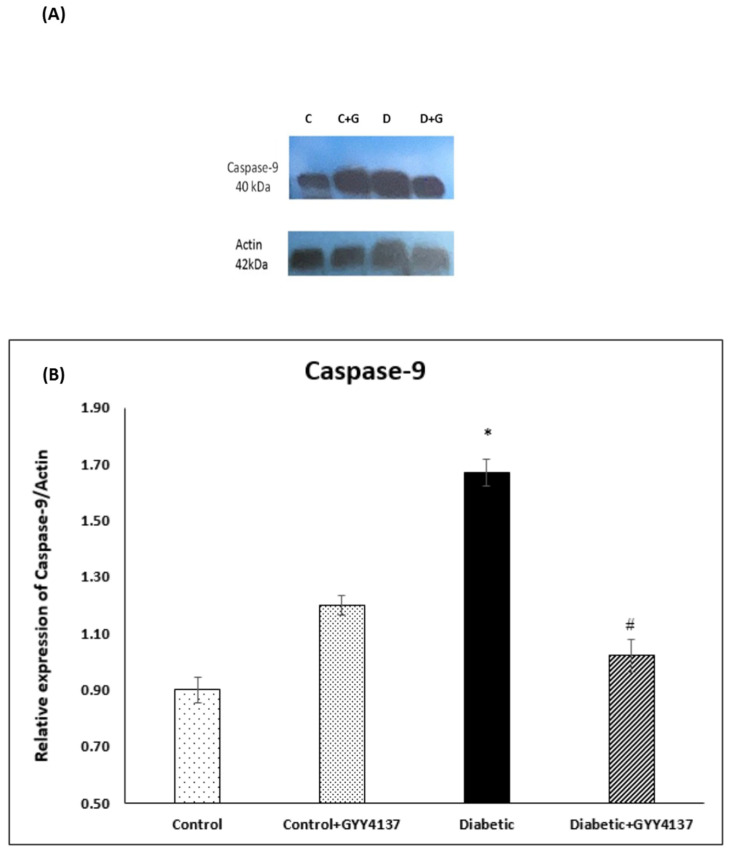
(**A**): Immunoblot of Caspase-9 in the lumbar region of the spinal cord. (**B**): Densitometry for Caspase-9 protein band expression. * *p* = 0.031; diabetic group vs. control groups. # *p* = 0.042; GYY4137-treated diabetic group vs. diabetic group. (one-way ANOVA followed by Bonferroni post-test). Values are mean ± SD, *n* = 5. C: control, D: diabetic, G: GYY4137.

**Figure 13 ijms-24-16650-f013:**
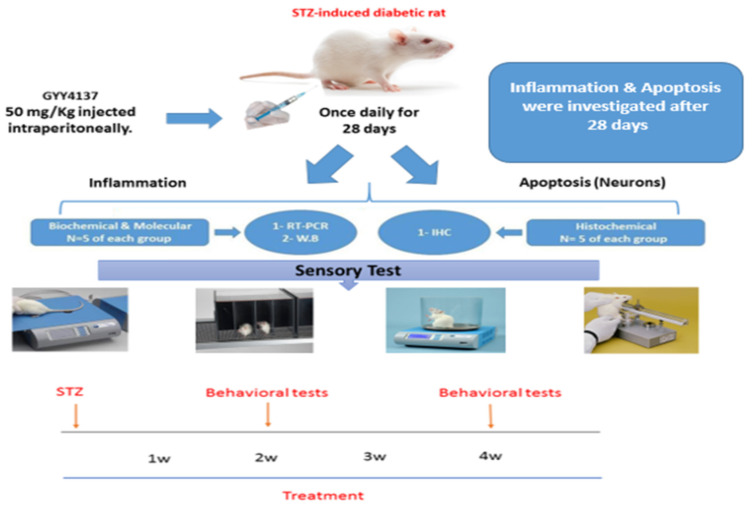
Schematic representation of the method outlined in the in vivo experiments. RT-PCR: Real time polymerase reaction, W.B: Western blot and IHC: Immunohistochemistry. Designed by Abdulaziz Shayea.

**Table 1 ijms-24-16650-t001:** Treatment groups.

Group	Treatment
1	Control (*n* = 10)
2	Control + GYY4137 (*n* = 10)
3	Diabetic (*n* = 15)
4	Diabetic + GYY4137 (*n* = 15)

**Table 2 ijms-24-16650-t002:** The sequences of primers used for real-time PCR.

*Bcl2*
Forward primer: 5′-CATGTGTGTGGAGAGCGTCAA-3′
Reverse primer: 5′-GCCGGTTCAGGTACTCAGTCA-3′
*Bax*
Forward primer:5′-GGGACGAACTGGACAGTAACAT-3′
Reverse primer:5′-GGAGTCTCACCCAACCACCCT-3′
β-actin
Forward primer: 5′-ATGAGCCCCAGCCTTCTCCAT-3′
Reverse primer: 5′-CCAGCCGAGCCACATCGCTC-3′

## Data Availability

Data will be made available on request.

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
