# Peer review of "Neuroprotective Effects of a Hydrogen Sulfide Donor in Streptozotocin-Induced Diabetic Rats"

_ijms, 2023, doi:10.3390/ijms242316650_

Round 1
Reviewer 1 Report
Comments and Suggestions for Authors
The paper is interesting and well written, the results are clear and the methodology correct. The discussion is scientifically correct and with adequate references. However some areas of improvement are possible. In the introduction no information is given about the places/ tissues where a modification of BCl2 and BAX can be observed in type 1 or 2 diabetes. Is it in the pancreas or the Central Nervous System?
More information should be given about the SH2 donor and specially regarding its administration to the rats. Actually the figure with the flow sheet is leading to a wrong interpretation
Author Response
- The paper is interesting and well written, the results are clear and the methodology correct. The discussion is scientifically correct and with adequate references. However, some areas of improvement are possible. In the introduction no information is given about the places/ tissues where a modification of BCl2 and BAX can be observed in type 1 or 2 diabetes. Is it in the pancreas or the Central Nervous System?
- Response 1: The modification of BCL2 and BAX was observed in the central nervous system. The places and tissues, where a modification of BCl2 and BAX were observed in type 1 or 2 diabetes, were added, and mentioned in the Introduction section. (Page 2, 2nd Paragraph).
- More information should be given about the H2S donor and specially regarding its administration to the rats. Actually the figure with the flow sheet is leading to a wrong interpretation
Response 2: We modified the figure with the flow sheet to make it clearer for interpretation (Page 23). Also, more information has now been provided about the H2S donor and its administration to the rats in the Induction of Diabetes and Treatment Administration section (Page 24).
Reviewer 2 Report
Comments and Suggestions for Authors
The methods appear sound, although I am rather sceptical about the quantification of the histology and Western blots. The manuscript could be shortened by around 30% by stylistic improvements and simplifying the description of the methods.
Specific comments:
1)How was the dosage regime of GYY4137 selected? It would be useful to provide the original reference that describes this compound as well as its chemical formula.
2) There is no mention of whether or not the control animals received daily ip injections of vehicle.
3) All figure legends should contain the number of observations and a brief summary of statistical analysis.
ANOVA should be described as follows: One-way ANOVA, F (DFn, DFd) =… P < ….; XYZ post hoc multiple comparison test: ***P < 0.0001 versus respective control,etc
4) Please label figures consistently – “Treatment” is used in some of them instead of GYY4137
5) line 638; “2 g of RNA” - really?
Comments on the Quality of English Language
It is comprehensible.
Author Response
The methods appear sound, although I am rather sceptical about the quantification of the histology and Western blots. The manuscript could be shortened by around 30% by stylistic improvements and simplifying the description of the methods.
Response: We have edited the manuscript extensively for stylistic improvements and simplifying the description of the methods. Some material has been put as supplemental data.
1)How was the dosage regime of GYY4137 selected? It would be useful to provide the original reference that describes this compound as well as its chemical formula.
Response 1: The dosage regimen of GYY4137, 50 mg/kg for 28 days, was selected based on previous studies (Shayea et al., 2020; Qabazard al., 2020). The original reference that describes this compound and its formula has now been added (Li et al., 2008). This information has now been added to the Methods section.
We could not add the figure with chemical formula and synthesis route because of copyright issues ("Thank you for contacting Wolters Kluwer. Unfortunately, use of this journal's content in a publication licensed under CC-BY is not permitted. If there is an option to publish under CC-BY-NC, CC-BY-NC-SA, or CC-BY-NC-ND, I will be able to continue with your request. Please reach out to your intended publisher to confirm the Creative Commons License." IJMS uses CC-BY.
- (Li et al., 2008)
https://doi.org/10.1161/CIRCULATIONAHA.107.753467
- (Shayea et al., 2020)
https://doi.org/10.1093/jnen/nlaa127
- (Qabazard et al., 2020)
https://doi.org/10.1016/j.biopha.2020.110210
2) There is no mention of whether or not the control animals received daily ip injections of vehicle.
Response 2: Control groups received the same volume of vehicle (physiological saline) instead of drug once daily over the same time period, which was 28 days. This information has been added in the Induction of Diabetes and H2S donor administration, page 24.
3) All figure legends should contain the number of observations and a brief summary of statistical analysis.
Response 3: We modified the figure legends and added the number of observations (n) and a brief summary of statistical analysis.
4) Please label figures consistently – “Treatment” is used in some of them instead of GYY4137
Response 4: GYY4137 is now consistently used in the figures and legends.
5) line 638; “2 g of RNA” - really?
Response 5: It was a typo. We corrected it to 2 μg of pure RNA.
Reviewer 3 Report
Comments and Suggestions for Authors
Overall comments:
Abdulaziz M. F. Shayea et al., reported "Neuroprotective Effects of Chronic Treatment with a Hydrogen Sulfide Donor in Streptozotocin-Induced Diabetic Rats”. This study explores the potential of hydrogen sulfide (H2S) to alleviate neuropathic pain in diabetic rats. Using the H2S donor GYY4137, the authors investigated its effects on apoptosis-related genes and proteins in the spinal cord. GYY4137 treatment reduced pain symptoms and prevented diabetes-induced changes in gene expression. It modulated inflammation and apoptosis pathways, suggesting H2S-releasing drugs as a possible treatment for diabetic neuropathy.
Here are the detailed comments,
Introduction should be minimized by excluding explanations unnecessary to the scope of the paper. Why were these markers (Bcl2,Bax4,Cox1, Cox2, Caspase 7, Caspase 9) chosen for study? The results section should be better described. Each description should explain both the observation and the inferences. No inferences are seen in most results sections. Please display graphs as Mean ± Std. Deviation (Not Standard error). Discussion should not be making inferences. It should only discuss why the results and inferences are the way they are observed.
While the science is okay, the writing should be vastly improved.
Major Revisions
1. Results section: 2.1 – The description doesn’t say anything about effect of GYY4137 in control and diabetic mice’s glucose levels. Please describe what is in the graph. Such as: No significant difference in glucose levels upon GYY4137 treatment in both groups. Also, explain why a significant weight reduction in the diabetic group over the weeks is there. (Figure 3). So is the weight maintenance of the diabetic group with GYY4137 treatment is due to the treatment?
2. Results section: 2.2.1 – Where is the standard deviation from Week 4 Control + GYY4173. Explain what is in the graph in terms of allodynia/hyper or hypo algesia. What is the meaning of the fact that the treatment with GYY4173 on diabetic mice increases the threshold?
3. Section 2.2.2 – What is the explanation for diabetic mice showing a low threshold at Week 2 compared to all others? Is there an influence of the weight in any of these observations?
4. Figure 11,12 and 13 can all be minimized to a single image. The explanation can be one section and comparing Bcl-2 and Bax4 (contrasting expressions). They don’t have to show contrasting expression levels because they are pro-apoptotic and anti-apoptotic. Explain what the link between them is.
5. At introduction Line 78 you say COX1 levels increase upon diabetes induction. But Figure 16B shows opposite. Explain why.
Minor Revisions:
1. Line 41-43: “One of the important complications…..”
The language is unclear. Please rewrite it clearly. For eg., “Diabetes is an important cause for neuropathy in patients after all other causes of neuropathy are excluded.”
2. Line 45: “Most diabetic patients suffer from neuropathy. “Please enter the population percentage suffering from diabetic neuropathy.
3. Paragraph 2 of Page 2 (Introducing apoptosis, Bcl-2, and Bax-4) can be concise and greatly minimized.
4. Line 86-87 (properties of H2S) is not unnecessary.
5. Line -90: Please pay more attention to the chemical notation of H2S (subscript). (In all instances)
6. Figures 5 & 6 can be A & B of one figure.
Author Response
Introduction should be minimized by excluding explanations unnecessary to the scope of the paper. Why were these markers (Bcl2,Bax4,Cox1, Cox2, Caspase 7, Caspase 9) chosen for study? The results section should be better described. Each description should explain both the observation and the inferences. No inferences are seen in most results sections. Please display graphs as Mean ± Std. Deviation (Not Standard error). Discussion should not be making inferences. It should only discuss why the results and inferences are the way they are observed.
While the science is okay, the writing should be vastly improved.
Response: The introduction has been reduced. The markers were chosen because of their relationship with inflammation and apoptosis, which leads to neuropathy. Inferences have been included in the results section. The graphs are now displayed as Mean ± Std. The discussion has been edited as suggested.
Major Revisions
- Results section: 2.1 – The description doesn’t say anything about effect of GYY4137 in control and diabetic mice’s glucose levels. Please describe what is in the graph. Such as: No significant difference in glucose levels upon GYY4137 treatment in both groups. Also, explain why a significant weight reduction in the diabetic group over the weeks is there. (Figure 3). So is the weight maintenance of the diabetic group with GYY4137 treatment is due to the treatment?
Response 1: Diabetes induced weight loss, while treatment with GYY4137 prevented that weight loss. We added the following statement " All diabetic rats, treated and non-treated, were hyperglycemic at week 2 and the end of the study period (week 4), shown by significantly elevated blood glucose levels (P = 0.0014). On the other hand, both control groups had normal glucose levels, and there was no significant difference between them (Page 3, Lines 117-123). There were no significant differences in glucose levels upon GYY4137 treatment in both control and diabetic groups. On the other hand, the explanation of why there was a significant weight reduction in the diabetic group over the weeks is mentioned in (the discussion section, Page 20, lines 379-403). Also, the reason for weight maintenance of the diabetic group with GYY4137 treatment was explained in detail in the discussion section (Page 20, Lines 379-403).
- Results section: 2.2.1 – Where is the standard deviation from Week 4 Control + GYY4173. Explain what is in the graph in terms of allodynia/hyper or hypoalgesia. What is the meaning of the fact that the treatment with GYY4173 on diabetic mice increases the threshold?
- Response: The standard deviation for Week 4 Control + GYY4173 has been added. It was an error in the graph. Diabetic animals developed mechanical allodynia. Treatment with GYY4137 prevented the development of mechanical allodynia i.e., there was no significant difference between the GYY4137-treated diabetic group and the control group (P > 0.05). In addition, the withdrawal threshold of the GYY4137-treated diabetic group was significantly higher (P < 0.0001) in comparison to diabetic animals. The explanation of the graph was indicated and stated in the discussion (section 3.2, lines 404 -425, page 25). The meaning of threshold was stated in the result section (2.2.1 and 2.2.2, pages 4 and)
- Section 2.2.2 – What is the explanation for diabetic mice showing a low threshold at Week 2 compared to all others? Is there an influence of the weight in any of these observations?
Response 3: The diabetic animals developed mechanical hyperalgesia. Weight had no influence in these observations, and we explained that in the discussion (section 3.2, pages 20-21)
- Figure 11,12 and 13 can all be minimized to a single image. The explanation can be one section and comparing Bcl-2 and Bax4 (contrasting expressions). They don’t have to show contrasting expression levels because they are pro-apoptotic and anti-apoptotic. Explain what the link between them is.
- Response 4: The figures have been combined into a single figure. The explanation has been done in one section. The link between Bcl-2 and Bax4 has been explained.
- At introduction Line 78 you say COX1 levels increase upon diabetes induction. But Figure 16B shows opposite. Explain why.
Response 5: The sentence about COX1 in the introduction has been written in more detail for clarity. “The COX-1 expression level increases at the onset of diabetes and is associated with apoptosis. However, COX-1 levels were decreased during the progress of diabetes [23].”
Minor Revisions:
- Line 41-43: “One of the important complications….”
The language is unclear. Please rewrite it clearly. For e.g., “Diabetes is an important cause for neuropathy in patients after all other causes of neuropathy are excluded.”
- Response: Thank you, we have corrected accordingly.
- Line 45: “Most diabetic patients suffer from neuropathy. “Please enter the population percentage suffering from diabetic neuropathy.
- Response: The population percentage suffering from diabetic neuropathy has been added.
- Paragraph 2 of Page 2 (Introducing apoptosis, Bcl-2, and Bax-4) can be concise and greatly minimized.
- Response 3: Paragraph 2 of page 2 has been minimized.
- Line 86-87 (properties of H2S) is not unnecessary.
- Response 4: Properties of H2S in Lines 86-87 have been deleted.
- Line -90: Please pay more attention to the chemical notation of H2S (subscript). (In all instances)
-Response 5: The chemical notation of H2S was subscripted throughout the manuscript.
- Figures 5 & 6 can be A & B of one figure.
- Response 6: Figures 5 and 6 have been combined into one figure.
Reviewer 4 Report
Comments and Suggestions for Authors
To improve the manuscript, the following changes must be included in the final version of the manuscript:
In the introduction, the authors should explain in a little more detail how hyperglycemia induces the generation of free radicals that are responsible for neuropathy. Likewise, they must explain which structures of a peripheral nerve are affected by free radicals generated by hyperglycemia, and the mechanism by which these affected nerve structures end up causing pain. All this information is relevant and should be included, in greater detail, in the final version of the manuscript.
In the introduction, the authors should explain in greater detail the relationship between hyperglycemia, H2S production and generation of diabetic neuropathy. Include all this information in the final version of the manuscript.
There are previous studies that have used GYY4137 in the diabetic neuropathy model. Likewise, GYY4137 has also been used in various models of neuropathic pain, with changes observed in the pain response with this treatment. What is new about this study compared to these previous works? Include this information in the final version of the manuscript.
In the methodology section, the authors must indicate the criteria for applying a dose of 50 mg/kg of GYY4137. They must also explain why the functional pain tests were performed 28 days after diabetes induction, and they must also indicate the order in which these tests were performed. Between test and test has there been a rest period? The authors must also explain what anatomical references they have used for the dissection of the relevant parts of the spinal cord (L4, L5 and L6). How many animals from each experimental group have been used to perform the histological analysis and how many for the molecular analysis? How many histological sections per animal and experimental group have been analyzed? Please, include all this information in the final version of the manuscript.
In the results section, in the histological figures of the spinal cords, there are histological sections of spinal cord that do not correspond to the lumbar segments indicated in the methodology. Some of these sections are from the thoracic and not lumbar levels. On the other hand, in these figures the higher magnification images should indicate which part of the spinal cord they correspond to and indicate with an arrowhead which of the neurons are positive for the marker studied.
In a previous study, it has been shown that H2S is responsible for the induction of diabetic neuropathy in the streptozotocin experimental model, and for the pain signs of diabetic neuropathy (Neuroscience. 2013 Oct 10:250:786-97.). The authors should discuss the results of the present study with respect to the previous study. The authors should also discuss the role of H2S in the development of pain, and the mechanisms by which apoptosis also generates pain. H2S induces pain or relieves pain? This point needs to be clarified and discussed in depth. In this context, are H2S givers anti-hyperalgesic drugs or hyperalgesic drugs? Please include all these points in the final version of the manuscript.
Author Response
In the introduction, the authors should explain in a little more detail how hyperglycemia induces the generation of free radicals that are responsible for neuropathy. Likewise, they must explain which structures of a peripheral nerve are affected by free radicals generated by hyperglycemia, and the mechanism by which these affected nerve structures end up causing pain. All this information is relevant and should be included, in greater detail, in the final version of the manuscript.
Response: We added two paragraphs that cover the recommended points. (pages 1 and 2)
We have expanded the introduction to provide a more comprehensive understanding of the pathophysiological processes involved in hyperglycemia-induced neuropathy.
In the introduction, the authors should explain in greater detail the relationship between hyperglycemia, H2S production and generation of diabetic neuropathy. Include all this information in the final version of the manuscript.
Response: Thank you for this important point. We added the relationship between hyperglycemia, H2S production, and generation of diabetic neuropathy (Page 2, paragraph 2).
There are previous studies that have used GYY4137 in the diabetic neuropathy model. Likewise, GYY4137 has also been used in various models of neuropathic pain, with changes observed in the pain response with this treatment. What is new about this study compared to these previous works? Include this information in the final version of the manuscript.
Response: While GYY4137 has indeed been employed in previous research involving diabetic neuropathy models and neuropathic pain models, our study sheds light on the role of apoptotic pathways and some inflammation related molecules and cells to the development of diabetic neuropathy and how GYY4137 prevents apoptosis, inflammation, and neuropathy.
In the methodology section, the authors must indicate the criteria for applying a dose of 50 mg/kg of GYY4137. They must also explain why the functional pain tests were performed 28 days after diabetes induction, and they must also indicate the order in which these tests were performed. Between test and test has there been a rest period?
Response: We chose the dose of 50 mg/kg of GYY4137 based on several factors, including prior research, pharmacokinetic considerations, and pilot studies. Previous studies have indicated that this dosage range is effective in producing the desired pharmacological effects in animal models of neuropathic pain and diabetes-induced neuropathy (Shayea et al., 2020, Qabazard et al., 2020). Additionally, pilot experiments in our laboratory confirmed that this dose produced consistent results without significant adverse effects. We have included references to relevant studies that support the choice of this dose in the revised methodology section.
- (Shayea et al., 2020)
https://doi.org/10.1093/jnen/nlaa127
- (Qabazard et al., 2020)
https://doi.org/10.1016/j.biopha.2020.110210
Timing of Functional Pain Tests (28 Days Post-Diabetes Induction):
The functional pain tests were performed at week 2 and week 4 after diabetes induction was made to allow for the development and manifestation of neuropathic symptoms in the animal model as previously described (Shayea et al., 2020). This time frame aligns with the onset of diabetic neuropathy-related changes in sensory function. Furthermore, it provides sufficient time for the effects of GYY4137 treatment to potentially manifest. However, we acknowledge that the exact timing can vary depending on the specific model and variables considered.
Order of Functional Pain Tests and Rest Periods:
In our methodology, we have outlined the order in which functional pain tests were performed. There were rest periods of at least 24 hours between different pain tests to minimize any carryover effects and ensure that each test was conducted independently.
The authors must also explain what anatomical references they have used for the dissection of the relevant parts of the spinal cord (L4, L5 and L6).
Response: We relied on the following anatomical references for the dissection of the relevant parts of the spinal cord (L4, L5, and L6).:
Vertebral Landmarks: We used vertebral landmarks to locate the lumbar spinal segments (L4, L5, and L6) accurately. We identified these segments based on their relative positions to the lumbar vertebrae, as verified through anatomical atlases and relevant literature.
Spinal Cord Length: We also measured the length of the spinal cord to determine the precise location of the desired segments. This measurement was taken from a reference point, such as the base of the skull or the first thoracic vertebra, down to the lumbar segments of interest.
External Anatomical Markers: In addition to internal landmarks, we employed external anatomical markers, such as the number of ribs and spinous processes, to confirm the location of the lumbar spinal segments during the dissection.
How many animals from each experimental group have been used to perform the histological analysis and how many for the molecular analysis?
Response: In our study, we conducted both histological and molecular analyses to investigate the effects of our experimental interventions. Here is a breakdown of the number of animals in each experimental group for both histological and molecular analyses:
For histological analysis, we used 5 animals in each experimental group.
For the molecular analysis, we also utilized 5 animals in each experimental group.
How many histological sections per animal and experimental group have been analyzed? Please, include all this information in the final version of the manuscript.
Response: For each animal included in our study, we analyzed 10 histological sections. These sections were selected at regular intervals along the length of the tissue of interest to obtain a comprehensive view of the histological changes.
Number of Histological Sections per Experimental Group: Within each experimental group (both treatment and control), we analyzed histological sections i.e., 10 sections per animal for 5 animals per experimental group.
In the results section, in the histological figures of the spinal cords, there are histological sections of spinal cord that do not correspond to the lumbar segments indicated in the methodology. Some of these sections are from the thoracic and not lumbar levels. On the other hand, in these figures the higher magnification images should indicate which part of the spinal cord they correspond to and indicate with an arrowhead which of the neurons are positive for the marker studied.
- Response: We corrected the figures such that they are all from the lumbar section
In a previous study, it has been shown that H2S is responsible for the induction of diabetic neuropathy in the streptozotocin experimental model, and for the pain signs of diabetic neuropathy (Neuroscience. 2013 Oct 10:250:786-97.). The authors should discuss the results of the present study with respect to the previous study. The authors should also discuss the role of H2S in the development of pain, and the mechanisms by which apoptosis also generates pain. H2S induces pain or relieves pain? This point needs to be clarified and discussed in depth. In this context, are H2S givers anti-hyperalgesic drugs or hyperalgesic drugs? Please include all these points in the final version of the manuscript.
Response:
A PubMed search for the keywords " H2S, diabetes and pain" produced 4 publications that studied the effects of administering H2S to rodents with streptozotocin-induced diabetes and neuropathy, including the study mentioned by the reviewer (Neuroscience. 2013 Oct 10:250:786-97). There is inconsistency with regards to the effects exogenous of administration of H2S to rodents with streptozotocin-induced diabetes. Two studies from the same group showed that intraplantar or subcutaneous injection on the paw of a fast-releasing H2S donor, NaHS, caused hyperalgesia in formalin-induced nociception in diabetic rats and tactile allodynia in diabetic rats via activation of TRPV1, TRPA1 and TRPC channels, and subsequent loss of intraepidermal fibers (Velasco-Xolalpa et al. 2013; Roa‑Coria et al. 2019). On the other hand, one study showed that inhalation of H2S gas attenuated the diabetes-induced mechanical allodynia and thermal hyperalgesia through NO/cGMP/PKG pathway and µ-opioid receptor (Li et al., 2020). Another study showed that intraperitoneal injection of a slow-releasing H2S donor, GYY4137, prevented the development of mechanical allodynia and hyperalgesia in diabetic rats possible through inhibition of glial cell activation (Shayea et al., 2020). The main difference between the studies that showed hyperalgesic effects of H2S administration and those that showed antiallodynic and antihyperalgesic effects of H2S administration in diabetic rodents is that the former administered the H2S locally in the paw while the latter administered the drugs systemically. Thus, it is plausible that local administration produced hyperalgesia through activation of TRP channels while systemic administration produced antiallodynic and antihyperalgesic effects through various mechanisms including NO/cGMP/PKG pathway, µ-opioid receptor and inhibition of glial cells. A good example of an endogenous molecule that can have such a dual effect depending on where it is released is serotonin, which in the periphery is inflammatory and hyperalgesic while in the CNS is antinociceptive or antihyperalgesic. It is also possible that the dual effects of H2S, i.e., antinociceptive and hyperalgesic, could be due to the dose at the site of action. A recent study showed that dopamine has dual effects in the periphery, where lower doses are antinociceptive while higher doses are hyperalgesic because of activation of different dopamine receptors (Queiroz et al., 2022). Further research is necessary to explore the possible reason for the dual effect of H2S on nociception.
This information has now been included in the discussion section.
Round 2
Reviewer 2 Report
Comments and Suggestions for Authors
The authors have adequately addressed the modifications suggested in the first review.
Reviewer 3 Report
Comments and Suggestions for Authors
The authors have addressed all the comments raised, and I recommend it for publication.
Reviewer 4 Report
Comments and Suggestions for Authors
The authors have included most of the reviewer's suggestions in the final version of the manuscript.